# Fusarium oxysporum infection-induced formation of agarwood (FOIFA): A rapid and efficient method for inducing the production of high quality agarwood

Zheng Zhang[1]☯, Meng Xiang-zhao[2]☯, Jiadong Ran[1]☯, Mei Gao[1], Ning-xiao Li[3], Yi-mian Ma[1], Ying Sun[1], Yuan Li[4]*

**1** Institute of Medicinal Plant Development, Chinese Academy of Medical Sciences and Chinese Peking Union Medical College, Beijing, China, **2** Institute of Biotechnology and Food Science, Hebei Academy of Agriculture and Forestry Sciences, Shijiazhuang, China, **3** Department of Plant Pathology and Environmental Microbiology, The Pennsylvania State University, University Park, PA, United States of America, **4** Institute of Plant Protection, Chinese Academy of Agricultural Sciences, Beijing, China

☯ These authors contributed equally to this work.
* liyuancaas@126.com

**Data Availability Statement:** All relevant data are within the manuscript and its Supporting Information files.

## Abstract

Agarwood, a non-wood product from the endangered *Aquilaria* and *Gyrinops* tress, is highly prized for its use in fragrances and medicines. The special formation process of agarwood is closely related to external injury and fungal infection. In this study, we demonstrate that infection of *Aquilaria sinensis* by *Fusarium oxysporum*, a soilborne fungus that causes vascular wilt diseases in diverse plants, induces agarwood formation. Based on these findings, an efficient method, termed *F. oxysporum* infection-induced formation of agarwood (FOIFA), was developed for the rapid production of quality agarwood. The agarwood formed in response to *F. oxysporum* infection was similar in structure and chemical composition to wild agarwood according to TLC (Thin-layer chromatography), HPLC (high performance liquid chromatography), and GC-MS (gas chromatography-mass spectrometry) analyses, except that the contents of alcohol-soluble extract, chromones, and essential oils (mainly sesquiterpenes) were higher in the formed agarwood.

## Introduction

Agarwood is one of non-timber forest products with strong antibacterial effects in nature, and is widely used in traditional medicine for sedation, pain relief and digestion. It is also culturally significant due to its use in incense ceremonies in Asia, the Middle East, and Europe [1]. In fact, agarwood is the most expensive wood globally [2]. The price of agarwood sold as wood chips ranges from US$ 6,000 to 20,000 per kilogram, depending on the quality, while essential oils extracted from agarwood can be worth up to US$ 30,000 per kilogram. The global market value of agarwood is estimated to be between US$6 billion and US$8 billion annually [3]. Due to the depletion of natural sources of agarwood driven by high demand for the product,

**Funding:** This work was supported by funds from the National Natural Science Foundation of China (81773844, 31000136) and the Beijing Municipal Natural Science Foundation (6102024). The funders had no role in study design, data collection and analysis, decision to publish, or preparation of the manuscript.

**Competing interests:** The authors have declared that no competing interests exist.

*Aquilaria* and *Gyrinops*, two important agarwood-producing genera, were listed as endangered species in 2002 [4].

Agarwood is a black resin that forms on the stems, branches and roots of injured *Aquilaria* and *Gyrinops* trees. The oleoresin of agarwood consists of a number of compounds, including sesquiterpenoids and phenylethyl chromone derivatives [5]. These compounds are not present in the healthy portion of *Aquilaria* wood, but rather in wounded or fungal-infected tissues [6, 7]. In general, wound formation and fungal infection are key factors in the formation of agarwood [8–10].

Many fungi isolated from natural agarwood, including *Epicoccum granulatum* [11], *Fusarium* sp., *Chaetomium globosum* [12], *Acremonium* sp. [13, 14], *Botryosphaeria dothidea* [15], *Lasiodiplodia theobromae* [16], and *F. solani* [17–19] have been found to be effective at inducing agarwood formation. The main advantage of deploying biological agents to induce agarwood formation is their ability to cause agarwood production progressively and continuously over the course of the fungal infection [20]. However, the fungal strains isolated from wild *Aquilaria* trees are mostly endophytes or saprophytes. Fungi that are highly virulent to *Aquilaria* might be more effective in inducing agarwood formation [10, 18]. *Fusarium solani*, which is highly virulent to *Aquilaria*, resulted in the formation of a resinous zone during the first week, and successfully induced agarwood formation after 3 months [18, 19].

In order to help meet the demand for agarwood production while protecting wild *Aquilaria* trees, this study had three objectives: i) to identify fungal pathogens of *Aquilaria* that can induce the production of agarwood; ii) to compare the chemical constituents of pathogen-inoculated agarwood and wild agarwood; and iii) to evaluate the yield and quality of agarwood formed by fungal infection.

## Materials and methods

### Isolation and identification of fungal isolates

Fungi were isolated from the stems of surface-sterilized symptomatic *Aquilaria sinensis* trees obtained from Lingshui County, Hainan Province, China. After seven days of incubation on Potato Dextrose Agar (PDA) plates at 25°C, individual fungal isolates were purified using the hyphal tip isolation technique [21]. Mycelia and conidia were observed under a light microscope (Olympus BX51, Tokyo, Japan) for morphological characterization. Spore counts were performed using a haemocytometer. Genomic DNA was extracted from fungal mycelia using a Precellys 24 tissue grinder and EZNA™ HP Fungal DNA Kit (Omega, USA), according to the manufacturers' instructions. PCR amplification of the ITS region was carried out using universal primers ITS1 (5′-TCCGTAGGTGAACCTGCGG-3′) and ITS4 (5′-TCCTCCGCTTA TTGATATGC-3′). The resulting amplicons were sequenced from both ends by Genelab (Invitrogen, Beijing). Sequence obtained were used as queries to search in Genbank. After the preliminary screening of the effect of all the obtained isolates and *A. sinensis*, an effective culture of the *F. oxysporum* isolate (AsFo20150101) was described in this study. A phylogenetic analysis using ITS sequences was performed with 1000 bootstraps in Mega 7.0 to understand the relationship between AsFo20150101 (GenBank Accession MW880244) and previously identified *Fusarium* isolates [22].

### Pathogenicity test and quantification of *F. oxysporum* in infected *A. sinensis*

Pathogenicity tests were conducted to determine whether the isolated *F. oxysporum* strain was the causal agent of vascular wilt in *A. sinensis*. Four-year-old *A. sinensis* trees planted in Xinyi County, Guangdong Province, China, were used. After sterilizing the trunk surface with 75% alcohol, a 4-mm diameter hole 20-mm deep was drilled in the stem approximately 300 mm above the ground to the trunk top, into which was injected 100 mL conidia suspension ($1\times10^6$/

mL) or ddH$_2$O via a sterilized transfusion set and by exploiting the transpiration pull, as previously described [16]. The deep hole was immediately sealed with sterilized paraffin wax to prevent microbes' invasion. The average data from three saplings (n = 3) in combination were statistically analyzed. After three weeks, 60-mm-long samples (with the bark removed) were collected at a height of 100 mm above the hole, and then 20-mm-long stem samples were used for histological observation as well as for qualitative and quantitative of *F. oxysporum*. The remaining 40-mm-long stem samples were immersed in liquid nitrogen and stored at −80˚C for chemical analysis of agarwood. To satisfy Koch's postulates, inoculated stem tissues were removed from seedlings exhibiting disease symptoms. After sterilizing with 70% ethanol, these stem tissues were then ground (1 gram per sample) using a Precellys 24 tissue grinder. Serial dilutions were plated on Komada plates (K$_2$HPO$_4$ (1 g), KCl (0.5 g), MgSO$_4$·7H$_2$O (0.5 g), FeNa-EDTA (0.01 g), L-asparagin (2 g), D-galactose (20 g), 1000 mL water) for *F. oxysporum* [23]. Genomic DNA of the resulting *F. oxysporum* strains was obtained and the internal transcribed spacer (ITS) region of ribosomal DNA was submitted to PCR using the *F. oxysporum* specific primers FOF1 (5′-ACATACCACTTGTTGCCTCG-3′) and FOR1 (5′-CGCCAATC AATTTGAGGAACG-3′), as previously described [24]. The PCR products of 340 bp were constructed to T vectors (pEasy-Blunt Simple Cloning Kit, TransGen Biotech), and then analyzed by the Shanghai bioengineering Co. LTD.

## Histological observation of *A. sinensis* stems infected by *F. oxysporum*

*A*. *sinensis* stems infected by *F. oxysporum* could be divided into three zones after three weeks, labelled as the N (Necrosis), B (Border), and H (Health) zones (Fig 3A). Subsequently, 60-μm thick sections were obtained from above zones via a freezing microtome. Transverse sections of each sample were observed and photographed under a light microscopy, as previously described [25].

## Chemical analysis of agarwood

**Thin-layer chromatography (TLC).** TLC analyses were carried out as previously reported [26] with slight modifications. Each finely ground sample (1 g) was dissolved in methanol at concentration 40 mg/ml. After TLC analysis (GF254, 10 × 20 cm, Merck) using CHCl$_3$:Et$_2$O (10:1, v/v) as the mobile phase, 5% (v/v) H$_2$SO$_4$–EtOH solution was used for staining. Wild agarwood samples (resinous stem wood) were obtained from Yanfeng Town, Haikou City, Hainan Province, China, which formed naturally due to strong winds.

**Content of alcohol-soluble extractive and essential oils.** The alcohol-soluble extractive and essential oils were extracted and quantified as previously described [26]. Two grams of sample were immersed in 100 mL of 95% (v/v) ethanol for 1 h, refluxed in a condenser for 1 h, cooled, and then filtered. Twenty-five milliliters of the filtrate were dried in an evaporating pan to a constant weight. After drying and cooling for 3 h, the percentage of alcohol soluble extractive was calculated with reference to the dried sample powder. The determination of essential oil in a sample is made by distilling a sample (50 g) with water (800 mL) in a volatile oil determination apparatus, collecting the distillate in a tube in which the aqueous portion of the distillate is automatically separated and returned to the distilling flask. The extracted essential oil was isolated and dried with anhydrous sodium sulfate, weighed and stored in sealed amber flasks at −20˚ C until analysis. Calculate the percentage of essential oil with reference to the dried sample powder. The experimental procedure was repeated twice.

**HPLC (high performance liquid chromatography) analysis.** The five reference standards (5S,6R,7S,8R)-2-(2-phenylethyl)-5,6,7,8-tetraydroxy-5,6,7,8-tetrahydro chromone (AH1), 6-hydroxy-2-2-(4′-methoxyphenylethyl)chromone (N), 6-hydroxy-2-(2-phenylethyl)

chromone (AH3), 6,7-dimethoxy-2-2-(4′-methoxyphenylethyl)chromone (AH8), and 6,7-dimethoxy-2-(2-phenylethyl)chromone (AH6) were prepared in our laboratory, and standard stock solutions were prepared by dissolving them in MeOH at suitable concentrations. The dried agarwood samples were powdered and screened through 40-mesh sieves. Accurately weighed powders (approximately 0.5 g) of each tested sample were mixed with 25 mL of MeOH and then subjected to ultrasonic extraction (ultrasonic cleaner, 53 kHz, Kudos, Shanghai, China) for 1 h. The extracted solution was adjusted to the original weight by adding MeOH. The final solution was filtered through a syringe filter (0.22 μm).

Five f chromones, AH1, N, AH3, AH8, and AH6 in agarwood were simultaneously determined using a Waters HPLC system, as previously reported [27]. The HPLC analysis was performed at 35˚C on a reverse-phase $C_{18}$ column using the water/acetonitrile gradient elution method. For detection, a photodiode array detector calibrated at 252 nm and 231 nm was used.

**GC-MS (gas chromatography-mass spectrometry) analysis.** The composition of essential oils was determined using an Agilent Technologies 7890B gas chromatograph system equipped with an HP-5 MS capillary column and a 5977A mass spectrometer equipped with an ion-trap detector. The application conditions of GC-MS and the identification process of sesquiterpene compounds were referred to our previous study [16]. The carrier gas was helium, at a flow rate of 1 mL min$^{-1}$. The injections were performed in splitless mode at 250˚C. 1 μL of essential oil solution in hexane (HPLCgrade) was injected. The operating parameters were the temperature program of 50˚C for 1 min, ramp of 10˚C min$^{-1}$ up to 155˚C (15 min), subsequent increase to 280˚C with an 8˚C min-1 heating ramp, and keeping at 280˚C for 10 min. The components were identified by comparison of their mass spectra with the NIST 2002 library data for the GC-MS system, as well as by comparison of their retention indices (RI) with the relevant literature data.19 The relative amount (RA) of each individual component of the essential oil was expressed as the percentage of the peak area relative to the total peak area. The RI value of each component was determined relative to the retention times (RT) of a series of $C_8$-$C_{40}$ n-alkanes with linear interpolation on the HP-5MS column.

## Results

### Morphological and molecular identification of *F. oxysporum* associated with vascular wilt in *A. sinensis*

The common disease symptoms of vascular wilt in *A. sinensis* include leaf yellowing, necrosis, and defoliation; these typically begin on the leaf margins of the lower leaves (Fig 1A–1C). As the disease progresses, the vascular system of the upper tap root and lower stem appears dark brown (DB) (Fig 1D), and eventually, the plant dies. The isolated fungal cultures formed white aerial mycelium and purple pigment (Fig 1E). The colony produced both microconidia and macroconidia (Fig 1F), which matched descriptions of *F. oxysporum* [28].

*F. oxysporum* was confirmed by sequence-based identification. The ITS sequence had over 99% similarity to nine *F. oxysporum* sequences available in GenBank. Phylogenetic analysis further confirmed that the isolated *F. oxysporum* was grouped with known formae speciales of *F. oxysporum* (Fig 2).

### Pathogenicity of *F.oxysporum* on *A.sinensis*

*A. sinensis* seedlings developed typical symptoms of vascular wilt such as leaf yellowing and wilting at 30 days post-inoculation (dpi) with *F. oxysporum*. Further, the vascular tissue of the stems also exhibited brown discolouration (Fig 3A). The stems were divided into three zones,

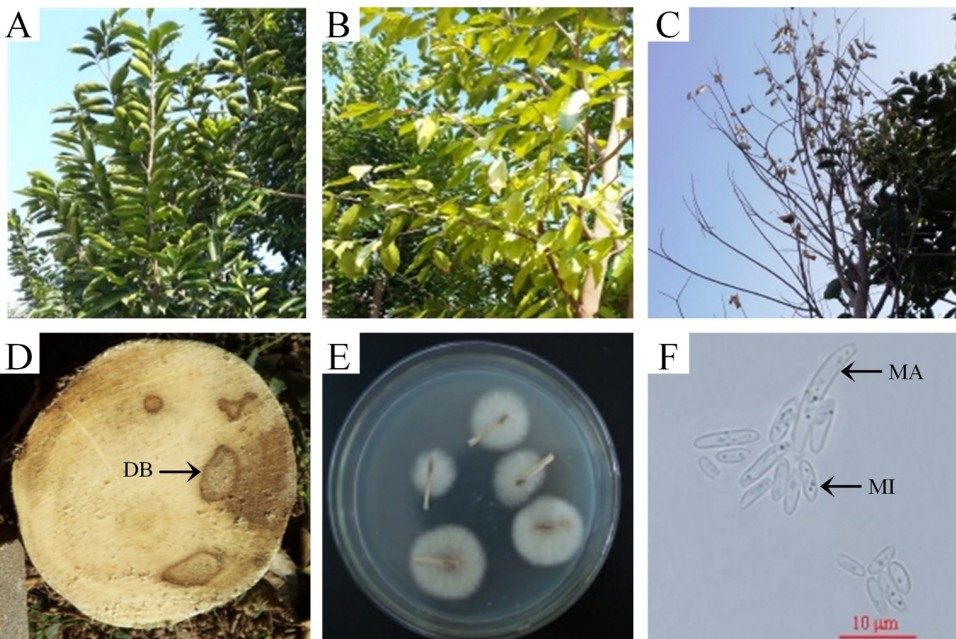

**Fig 1. Symptoms of vascular wilt in *A. sinensis* trees and morphological characteristics of *F. oxysporum* isolated from an infected tree.** (A) Healthy tree branches. (B,C) Infected tree with yellowing, defoliation. (D) Symptoms after infection. *A. sinensis* stems are dark brown (DB) when infected by pathogens. (E) Fungal colonies formed on PDA after three days. (F) Microconidia (MI) and macroconidia (MA) of a 10-day-old *F. oxysporum* culture on PDA.

N (Necrosis), B (Border) and H (Health) zones (Fig 3A). The H zone comprised white wood and vascular occlusions were absent in the vessels (Fig 3B). In zone B, brown or dark-brown resin filled in ray and axial parenchyma cells, as well as the vessels and wood fibres (Fig 3C). These features in the B zone are similar to those observed in natural heartwood, leading us to speculate that the B zone may contain agarwood substances (Fig 3E and 3F). In the N zone, ray and axial parenchyma cells were necrotic and fragmented, and fungal hyphae were found in the vessel lumens of the secondary xylem (Fig 3D). In summary, both the B and N zones are areas infected by *F. oxysporum*.

To fulfil Koch's postulates, the pathogens in the different zones were isolated on selective media for *F. oxysporum*. The recovered isolates were morphologically identical to the inoculum. The *F. oxysporum*-specific gene regions were amplified from all recovered isolates, and their identity as *F. oxysporum* was further confirmed (Fig 4A). The colony forming unit (CFU/g) of the samples from the N, B, and H zones were isolated; the CFU/g decreased sharply from the N zone to the B zone, and then to the H zone. The largest number of *F. oxysporum* (13,700 CFU/g) was detected in the N zone and decreased to 3566 CFU/g in the B zone. In contrast, there was no *F. oxysporum* detected in the H zone (Fig 4B). The qualitative and quantitative results for *F. oxysporum* were consistent with the light microscope observations.

## *F. oxysporum* induced agarwood formation

The TLC results revealed identical spots for the methanol extracts from infection zone, also called brown zone (B) which shown in Fig 1D and wild agarwood (W) which selected as a control, as visualized under UV254 and 365 nm (Fig 5A and 5B). In addition, B exhibited similar colour characteristics to W after colouring with 5% $H_2SO_4$–EtOH solution (Fig 5C). Further, there were more constituents and higher contents of non-polar compounds of agarwood from

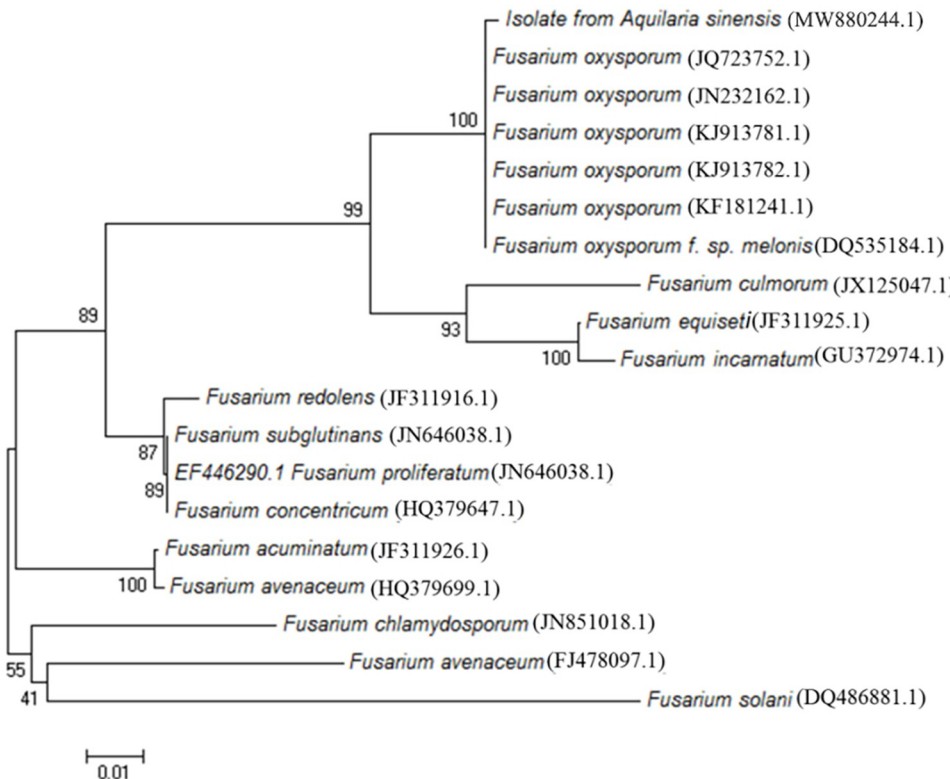

**Fig 2. Phylogenetic analysis of the isolated *F. oxysporum* with a selection of *Fusarium* species.**

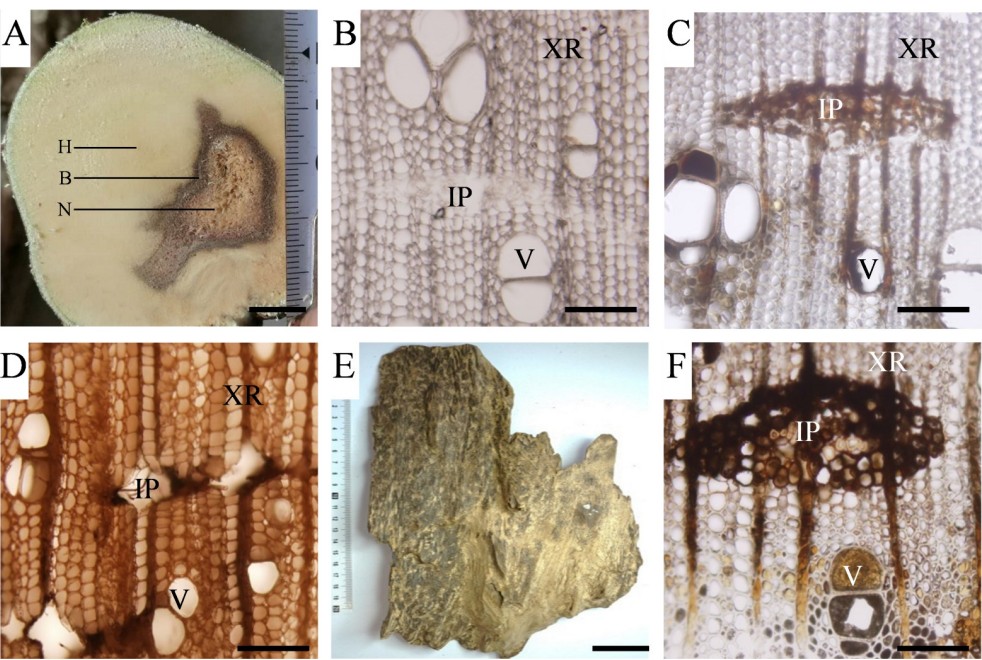

**Fig 3. Histological observation of stems inoculated with *F. oxysporum* and wild agarwood sample of *A. sinensis*.**
(A) Tangential section of *A. sinensis* stem inoculated with *F. oxysporum* 30 days post-inoculation, Bars = 10 mm. (B-D) Light microscope images of the transverse sections of the N (Necrosis), B (Border) and H (Health) zones, respectively, Bars = 50 μm. (E) Wild agarwood sample, Bars = 50 mm. (F) Transverse section of wild agarwood, Bars = 50 μm. IP: interxylary phloem, XR: xylem ray, V: vessel.

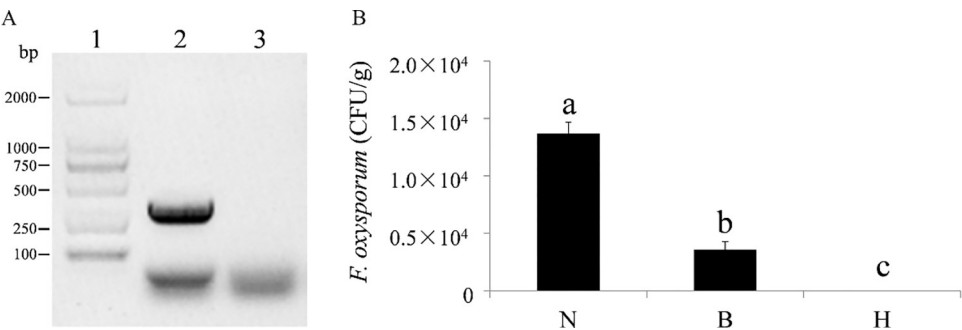

**Fig 4. Identification and quantification of *F. oxysporum* in stems of *A. sinensis* after one-month inoculation.** (A) Agarose gel electrophoresis of PCR products using the *F. oxysporum*-specific primers FOF1 and FOR1, Lane 1: DL2000 DNA marker (Takara, Japan), Lane 2: *F. oxysporum*, Lane 3: negative control. (B) Distribution of the colony forming unit per one gram (CFU/g) of *F. oxysporum* in the N (Necrosis), B (Border) and H (Health) zones.

B compared to agarwood from W. The above experimental results show that the quality of the agarwood induced by *F. oxysporum* was similar and actually better than that of wild agarwood. However, there was no spot detected in healthy wood (H).

Extracts from the H and B zones of the infected stems and from wild agarwood were analysed to determine the contents of alcohol-soluble extractive and essential oils. The yield of alcohol-soluble extractive of B was 15.60%, similar to that of W (14.32%). The yield of essential oils in B was 0.31%, which was much higher than that of W (0.17%). The contents of alcohol-soluble extractive and essential oils in H were 4.88% and 0.03%, respectively, much less than B and W (Fig 6).

## Similar chromone profiles of B and W

Chromones extracted from H, B, and W were also analysed via HPLC to determine the contents of individual chromones in these zones (Fig 7). Comparison of each peak's retention time with that of the five chromone standards in the same chromatographic system revealed

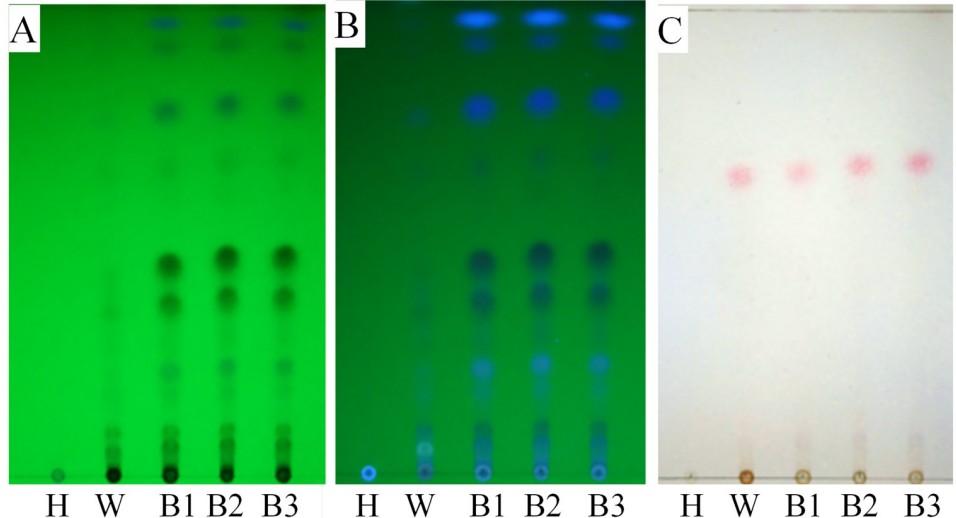

**Fig 5. Thin-layer chromatography (TLC) chromatogram of methanol extracts.** TLC chromatogram visualized under UV254 nm (A) and UV365 nm (B). (C) Stained and immobilized TLC chromatogram.

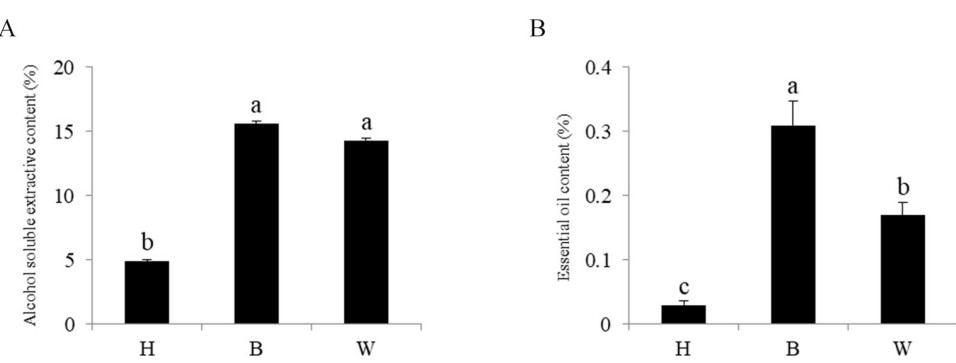

**Fig 6. The contents of alcohol-soluble extractive and essential oils in the brown zone (B), wild agarwood (W), and healthy zone (H).** (A) Alcohol soluble extractive content; (B) essential oil content.

that all five chromones were present in the B and W samples, but their contents in W were significantly lower than those in B (Fig 7). B had the highest chromones contents, with the major constituents being AH6 and AH1 (AH1 (4.2 mg/g), N (0.6 mg/g), AH3 (2.4 mg/g), AH8 (1.2 mg/g), and AH6 (6.2 mg/g), respectively). The predominant chromone in W was AH1 (AH1 (1.0 mg/g), N(0.1 mg/g), AH3 (0.1 mg/g), AH8 (0.2 mg/g), and AH6 (0.3 mg/g), respectively). In contrast, almost no chromones were detected in H (Figs 7 and 8).

## The composition of essential oils in B was similar to that in W

Although B had a higher essential oils content than W (Fig 6B), B had similar components to W (Fig 9 and Table 1). Both these oils were rich in sesquiterpenes, accounting for 76.77% in B and 75.98% in W. Fifty-two components were identified in B, and most of constituents were sesquiterpenes compounds such as α-Eudesmol (8.37%), Aristolone (8.14%), cis-Z-α-Bisabolene epoxide (7.68%), and 2,2,6-Trimethyl-1-[(1E)-3-methyl-1,3-butadienyl]-5-methylene-7-oxabicyclo[4.1.0]heptane (5.21%). Fifty-four components were identified in the essential oils of W, where the predominant compounds were sesquiterpenes, including cis-Z-α-Bisabolene epoxide (9.07%), α-Eudesmol (8.24%), and Guai-1(10)-en-11-ol (7.09%). The essential oils of H had significantly different components to those of W and B. The essential oils of H contained a small amount of sesquiterpenes (8.14%) which was richly distributed in the oils from W and B. However, H was rich in alkanes, accounting for 64.95% of the total oil content (Table 1).

## Discussion

Agarwood has been used for many high-value products as incense, perfumes (essential oils), in medicine and religious ceremonies, but the formation of its resin is very rare and slow under natural conditions. For the sustainable development of the agarwood industry, many agarwood-producing countries (Cambodia, China, Indonesia, Malaysia, Thailand, and Vietnam) have been developed physical, chemical, and biological induction techniques. Among these three techniques, biological (fungal inoculation) induction is faster than physical induction and safer than chemical induction [29]. Use of fungal inoculum is highly prioritized among all type of fungi due to the regular growth of fungal mycelia which continuously spreads in the plant, resulting in wound development and high-yield agarwood formation [16]. Previous study has shown that crude extracts of *Fusarium* could not lead to the formation or accumulation of sesquiterpenes and 2–2 phenythylchromone in shoot culture of *Aquilaria* specie plants, whereas methyl jasmonate does [30]. This indicates that the process of agarwood formation

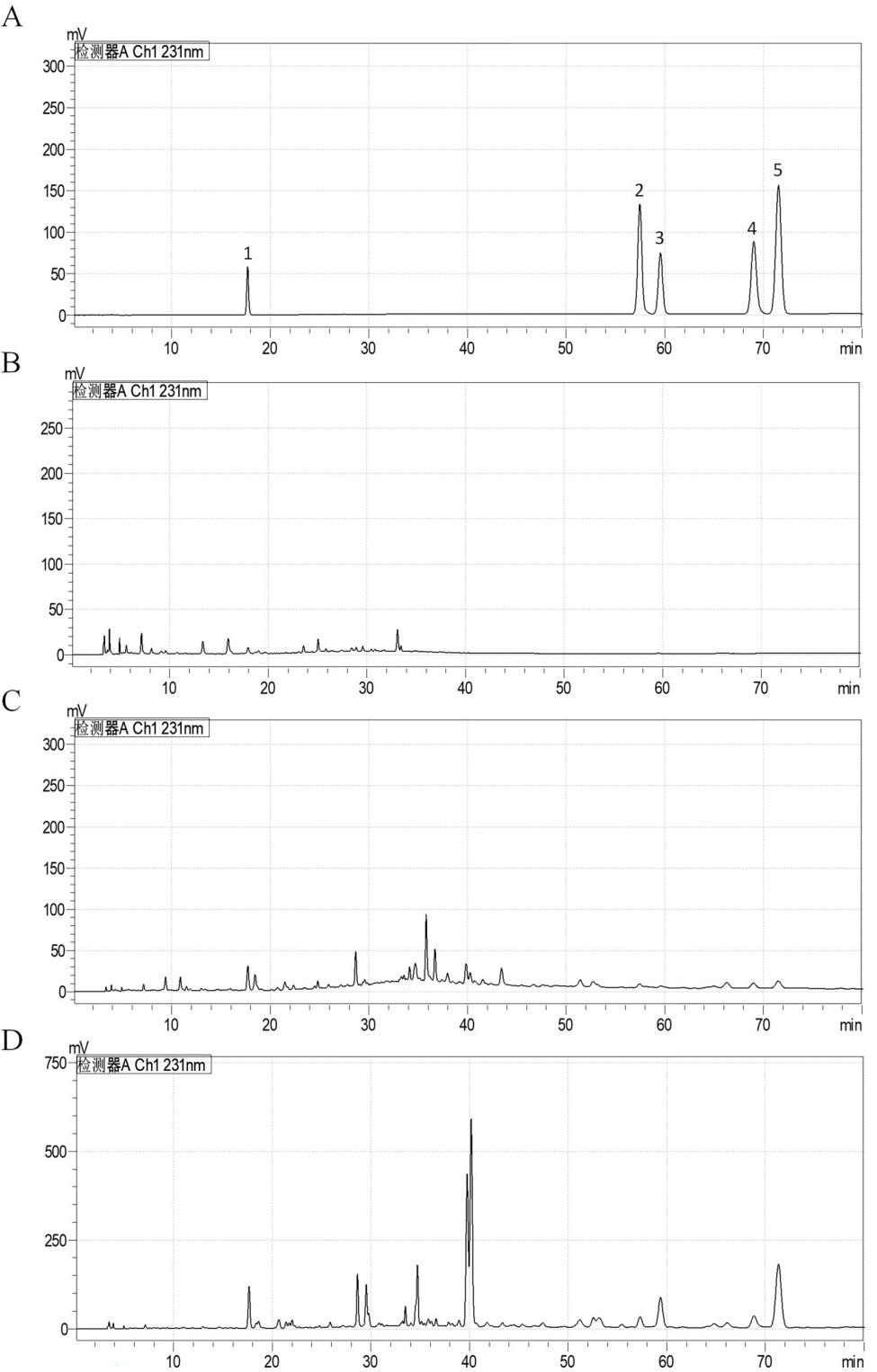

**Fig 7. HPLC chromatograms of five chromones in the brown zone (B), wild agarwood (W), and healthy zone (H).**
The standard mixtures of (1) AH1, (2) N, (3) AH3, (4) AH8, and (5) AH6 are shown in (A). The presence of these
chromones in H, B, and W are shown in (B-D).

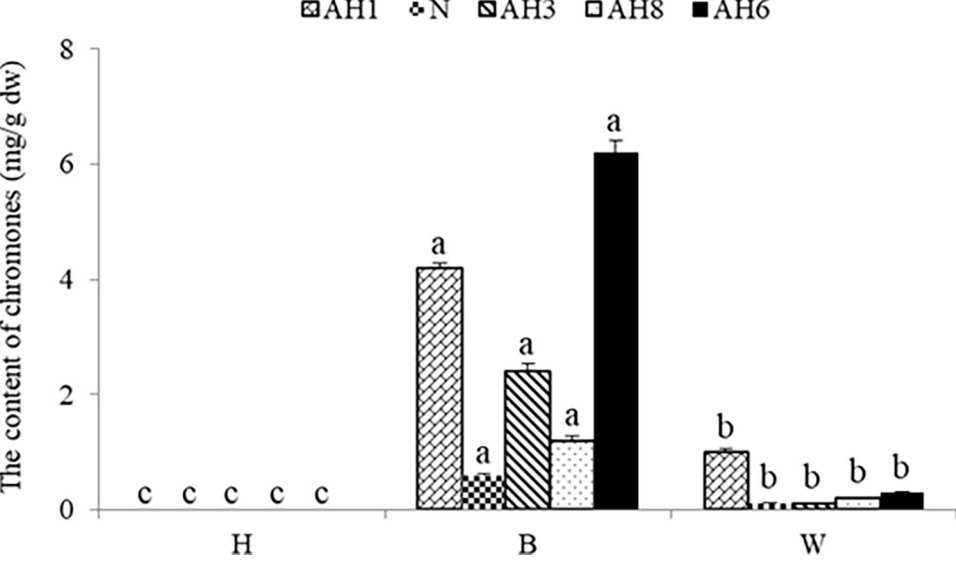

**Fig 8. The contents of five chromones in the brown zone (B), wild agarwood (W), and healthy zone (H).**

induced by *Fusarium* is not caused by itself, but may be caused by the secretion of *Fusarium* generated to stimulate the defense response of plant, promote the production of endogenous hormones in *Aquilaria*, and then initiate the synthesis of agarwood components. In this study, the pathogen associated with the disease symptoms of vascular wilt on *A. sinensis* was isolated and identified as *F. oxysporum* based on its cultural and morphological characteristics, the ITS sequencing results, and pathogenicity assays. The disease severity indices showed that after inoculation with *F. oxysporum*, seedlings exhibited the typical symptoms of vascular wilt; in particular, the plants began to produce agarwood in the B zones of the inoculated stems. Thus, the results indicated that *F. oxysporum* effectively elicited agarwood production in the stem of *A. sinensis*. The agarwood from the stem produced by fungal inoculation had a similar structural and chemical profile to wild agarwood product naturally produced from the stem. Interestingly, the agarwood produced by inoculation had higher alcohol soluble extract contents, chromones contents, and essential oils (mainly sesquiterpenes) contents than the wild agarwood product.

Although many endophytic strains have been shown to induce agarwood production through artificial induction experiments, there are fewer studies of natural pathogens from agarwood trees. Cowan et al. [31] reported that under pathological conditions, plants have an unlimited ability to synthesize terpenes, aromatic compounds, and their oxygen-substituted derivatives, which inhibit the growth of the infecting agent. In the current study, for the first time, 52 components were identified from *A. sinensis* infected by *F. oxysporum*, and these were of a similar composition to that of wild agarwood, especially sesquiterpenes and aromatic constituents. These results indicate that *F. oxysporum*, the causal agent of vascular wilt on *A. sinensis*, is a promising fungal isolate that deserves further study. Further, there is a potential for scale up to a commercial level for production of agarwood and its essential oils.

Previously, we developed a novel method for the formation of agarwood named the whole-tree agarwood-inducing technique (Agar-Wit), in which, chemical inducers are injected into the xylem vessels of *Aquilaria* wood, leading to the formation of resin deposits in 6 months [1, 16]. The alcohol soluble extractive contents of the ten samples obtained by the Agar-Wit with ten different agarwood inducers ranged from 11.60% to 18.08%, all surpassing the required

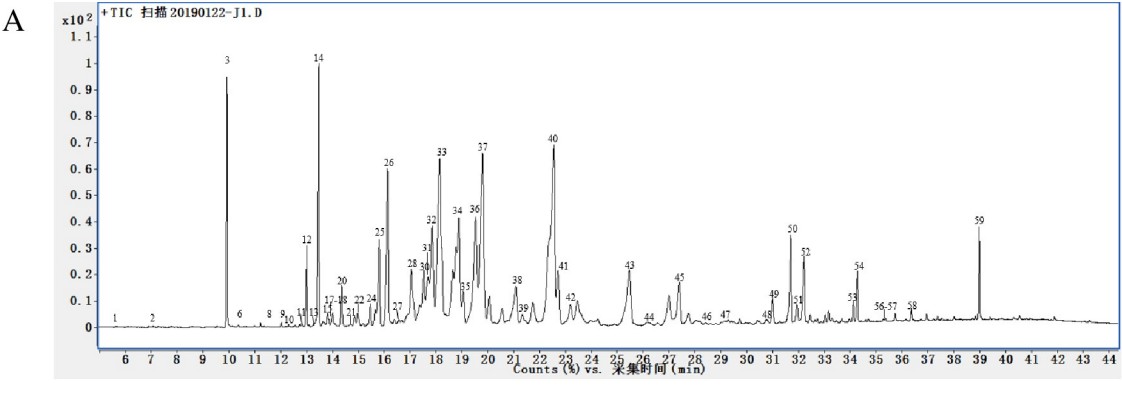

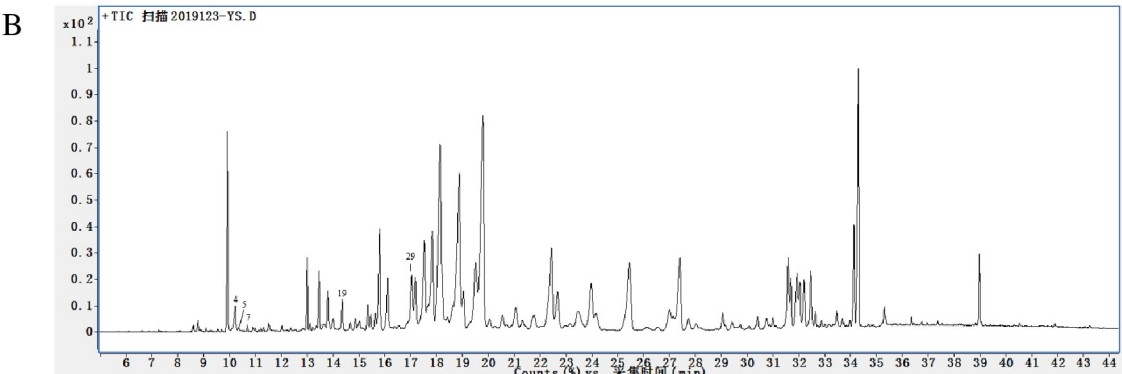

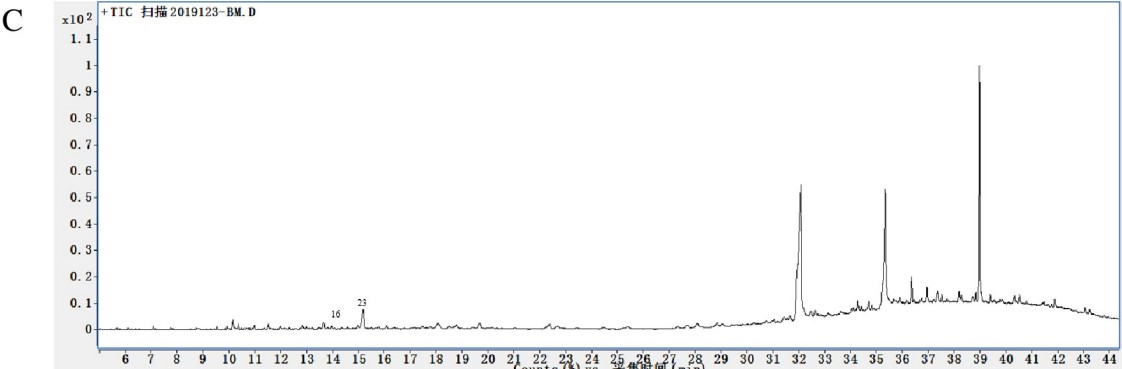

**Fig 9. GC chromatograms of the three essential oils in the brown zone (B), wild agarwood (W), and healthy zone (H).**

10% standard, and similar to that of wild samples (10.56% and 19.30%). In this study, another faster technique for agarwood induction, known as *F. oxysporum* infection-induced formation of agarwood (FOIFA), was invented. After three weeks of fungal inoculation, the yield of alcohol-soluble extractive of B was 15.60%, exceeding that of W (14.32%). Furthermore, the yield of total five chromones (AH1, N, AH3,AH8, and AH6) and essential oils in B was 14.6 mg/g and 0.31%, which were much higher than that of W (1.7 mg/g and 0.17%). Taken together, the agarwood formed three weeks after FOIFA treatment had higher quality than the wild agarwood. We suggest that, using our FOIFA, qualified agarwood may be produced from *Aquilaria* trees in one month, and high-quality agarwood may be obtainable if prolonging the agarwood-formation time appropriately. These results confirmed that FOIFA was a fast and

**Table 1. Chemical composition and relative amounts of essential oils in the brown zone (B), wild agarwood (W), and healthy zone (H).**

| No.[a] | Compound | RI[b] | RI[c] | Relative amount / %[d] | | |
|---|---|---|---|---|---|---|
| | | | | B | W | H |
| | **Aromatic compounds** | | | **2.63** | **3.13** | **1.56** |
| 1 | Benzaldehyde | 960 | 977 | 0.01 | 0.02 | - |
| 2 | Benzeneacetaldehyde | 1044 | 1049 | 0.02 | 0.01 | 0.12 |
| 3 | 2-Butanone, 4-phenyl- | 1248 | 1255 | 2.52 | 2.44 | 0.24 |
| 5 | Benzaldehyde, 4-propyl- | 1278 | - | - | 0.09 | 0.23 |
| 7 | Naphthalene, 2-methyl- | 1302 | 1300 | - | 0.08 | - |
| 8 | 3-Buten-2-one, 4-phenyl- | 1363 | 1346 | 0.01 | - | 0.30 |
| 16 | Phenol, 2,4-bis(1,1-dimethylethyl)- | MS | 1513 | - | - | 0.28 |
| 47 | 1,2-Benzenedicarboxylic acid, bis(2-methylpropyl) ester | MS | 1870 | 0.07 | 0.48 | 0.39 |
| | **Sesquiterpenes** | | | **76.77** | **75.98** | **8.14** |
| 10 | α-Guaiene | 1416 | 1413 | 0.03 | 0.08 | - |
| 11 | (+)-Longifolene | MS | 1428 | 0.09 | 0.06 | - |
| 12 | Humulene | 1462 | 1468 | 1.03 | 0.98 | - |
| 13 | Azulene, 1,2,3,3a,4,5,6,7-octahydro-1,4-dimethyl-7-(1-methylethenyl)-, [1R-(1α,3aβ,4α,7β)]- | 1479 | 1479 | 0.11 | 0.07 | - |
| 14 | α-Selinene | 1490 | 1490 | 3.87 | 0.89 | 0.20 |
| 15 | 2-Butanone, 4-(4-methoxyphenyl)- | 1504 | 1473 | 0.35 | 0.69 | - |
| 17 | 2H-3,9a-Methano-1-benzoxepin, octahydro-2,2,5a,9-tetramethyl-, [3R-(3.alpha.,5a.alpha.,9.alpha.,9a.alpha.)]- | MS | 1501 | 0.35 | 0.25 | - |
| 18 | 4,6,6-Trimethyl-2-(3-methylbuta-1,3-dienyl)-3-oxatricyclo[5.1.0.0(2,4)]octane | 1515 | - | 0.06 | 0.02 | - |
| 19 | Cedranoxide, 8,14- | MS | 1539 | - | 0.03 | - |
| 20 | Isolongifolan-8-ol | 1520 | 1523 | 0.70 | 0.51 | - |
| 21 | (-)-Spathulenol | MS | 1551 | 0.04 | 0.01 | - |
| 22 | Cyclohexanemethanol, 4-ethenyl-.alpha.,.alpha.,4-trimethyl-3-(1-methylethenyl)-, [1R-(1.alpha.,3.alpha.,4.beta.)]- | 1552 | 1542 | 0.36 | 0.29 | - |
| 23 | 2,6-Dimethyl-10-methylene-12-oxatricyclo[.0(1,6)]tridec-2-ene | 1576 | 1579 | - | - | 2.73 |
| 24 | 5β,7βH,10α-Eudesm-11-en-1α-ol | MS | 1588 | 0.45 | 0.32 | - |
| 25 | Caryophyllene oxide | 1596 | 1595 | 2.00 | 2.29 | - |
| 26 | cedrenol | 1604 | 1610 | 3.88 | 1.38 | - |
| 27 | Isoaromadendrene epoxide | 1606 | 1612 | 0.21 | 0.17 | - |
| 28 | γ-Eudesmol | 1625 | 1632 | 2.60 | 1.88 | - |
| 29 | Octahydro-2,2,4,7α-tetramethyl-1,3α-ethano(1H)inden-4-ol | 1630 | 1640 | -[e] | 1.50 | - |
| 30 | Hinesol | 1638 | 1638 | 1.82 | 3.00 | - |
| 31 | Agarospirol | 1645 | 1643 | 1.74 | - | - |
| 32 | (-)-Aristolene | 1647 | 1657 | 3.73 | 4.10 | - |
| 33 | α-Eudesmol | 1652 | 1660 | 8.37 | 8.24 | 1.10 |
| 34 | Guai-1(10)-en-11-ol | 1669 | 1669 | 3.26 | 7.09 | - |
| 35 | Aromadendrene oxide-(2) | 1705 | 1704 | 0.99 | 1.14 | - |
| 36 | 2,2,6-Trimethyl-1-[(1E)-3-methyl-1,3-butadienyl]-5-methylene-7-oxabicyclo[4.1.0]heptane | MS | 1710 | 5.21 | 3.41 | - |
| 37 | cis-Z-α-Bisabolene epoxide | 1704 | 1713 | 7.68 | 9.07 | - |
| 38 | 6-Ethenylhexahydro-6-methyl-3-methylene-7-(1-methylethyl)-2(3H)-benzofuranone | MS | 1740 | 1.02 | 0.46 | - |
| 39 | Longifolenaldehyde | MS | 1741 | 0.53 | 0.50 | - |
| 40 | Aristolone | 1762 | 1761 | 8.14 | 4.05 | - |
| 41 | (-)-Isolongifolol | 1781 | 1771 | 1.99 | 1.56 | 0.29 |
| 42 | 2(1H)Naphthalenone, 3,5,6,7,8,8a-hexahydro-4,8a-dimethyl-6-(1-methylethenyl)- | MS | 1773 | 1.05 | 0.28 | - |
| 43 | 2(3H)-Naphthalenone, 4,4a,5,6,7,8-hexahydro-4a,5-dimethyl-3-(1-methylethylidene)-, (4ar-cis)- | 1801 | 1828 | 3.32 | 3.75 | - |
| 44 | Diepicedrene-1-oxide | MS | - | 0.21 | 0.20 | - |
| 45 | Acetic acid, 3-hydroxy-6-isopropenyl-4,8a-dimethyl-1,2,3,5,6,7,8,8a-octahydronaphthalen-2-yl ester | MS | 1847 | 1.75 | 2.97 | - |
| 48 | 7,9-Di-tert-butyl-1-oxaspiro(4,5)deca-6,9-diene-2,8-dione | MS | 1919 | 0.09 | 0.35 | 0.19 |

*(Continued)*

**Table 1.** (Continued)

| No.[a] | Compound | RI[b] | RI[c] | Relative amount / %[d] | | |
|---|---|---|---|---|---|---|
| | | | | B | W | H |
| 49 | 1,5-Dimethyl-3-hydroxy-8-(1-methylene-2-hydroxyethyl-1)-bicyclo[]dec-5-ene | MS | 1969 | 1.87 | 0.98 | 0.51 |
| 52 | Eudesma-5,11(13)-dien-8,12-olide | MS | 1987 | 1.53 | 1.17 | - |
| 53 | 3-Oxo-10(14)-epoxyguai-11(13)-en-6,12-olide | MS | - | 0.33 | 1.62 | - |
| 54 | Cycloisolongifolene, 8,9-dehydro-9-formyl- | MS | - | 0.77 | 4.36 | - |
| | **Alkanes** | | | **1.97** | **4.80** | **64.95** |
| 4 | Nonanoic acid | 1268 | 1272 | - | 0.64 | 0.71 |
| 6 | Pentadecane | 1280 | - | 0.02 | - | 0.23 |
| 9 | Tetradecane | MS | - | 0.06 | 0.08 | 0.27 |
| 46 | Pentadecanoic acid | MS | 1869 | 0.02 | 0.03 | 0.64 |
| 51 | n-Hexadecanoic acid | 1967 | 1964 | 0.03 | 1.09 | 17.15 |
| 55 | n-Tetracosanol-1 | 2082 | - | 0.03 | 0.04 | 0.36 |
| 56 | 9,12-Octadecadienoic acid (Z,Z)- | MS | 2134 | 0.06 | 0.09 | 0.21 |
| 57 | Oleic Acid | 2138 | 2153 | 0.07 | 0.34 | 17.55 |
| 58 | 9-Tricosene, (Z)- | MS | 2298 | 0.16 | 0.04 | 2.94 |
| 59 | Phenol, 2,2'-methylenebis[6-(1,1-dimethylethyl)-4-methyl- | MS | 2398 | 1.02 | 0.84 | 15.37 |
| 50 | Dibutyl phthalate | MS | 1959 | 0.51 | 1.61 | 9.53 |
| TOTAL | | | | 81.37 | 83.92 | 74.65 |

Component numbers correspond to those in Table 1 and GC conditions are described in the Methods section of this paper.

[a] Order of elution is given on HP-5MS

[b] RI indicates reported in the literature for the HP-5MS column

[c] RI indicates the retention indices that were calculated against $C_8$-$C_{40}$ n-alkanes on the VF-5MS column

[d] relative amount determined as the peak area relative to the total peak area

[e] not detected.

efficient method for inducing the production of high quality agarwood. The popularization and application of FOIFA in agarwood-producing countries (Cambodia, China, Indonesia, Malaysia, Thailand, and Vietnam) may consecutively supply more agarwood and essential oils to the international markets. This may not only satisfy the high demand for wild agarwood, but also conserve and protect wild *Aquilaria* trees.

## Conclusions

In this study, a pathogenic fungus, *Fusarium oxysporum*, which can induce the production of high quality agarwood was found and identified. TLC (Thin-layer chromatography), HPLC (high performance liquid chromatography), and GC-MS (gas chromatography-mass spectrometry) analyses showed that the agarwood formed in response to *F. oxysporum* infection was similar in structure and chemical composition to wild agarwood, and the contents of alcohol-soluble extract, chromones, and essential oils (mainly sesquiterpenes) were higher in the formed agarwood. Based on these findings, an efficient method, termed *F. oxysporum* infection-induced formation of agarwood (FOIFA), was developed for the rapid production of quality agarwood.

## Supporting information

**S1 Raw images.**
(PDF)

## Author Contributions

**Formal analysis:** Jiadong Ran, Mei Gao, Yi-mian Ma, Ying Sun.

**Methodology:** Ning-xiao Li.

**Project administration:** Yuan Li.

**Writing – original draft:** Zheng Zhang.

**Writing – review & editing:** Meng Xiang-zhao.

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
