## [Decision Letter · Decision Letter 0]

25 Feb 2022

PONE-D-22-01806Fusarium oxysporum infection-induced formation of agarwood (FOIFA): a novel method for inducing the production of high quality agarwoodPLOS ONE

Dear Dr.xiangzhao,

Thank you for submitting your manuscript to PLOS ONE. After careful consideration, we feel that it has merit but does not fully meet PLOS ONE’s publication criteria as it currently stands. Therefore, we invite you to submit a revised version of the manuscript that addresses the points raised during the review process. After reviewing the manuscript and by going through reviewers comments I feel that the authors should clearly mention the innovativeness and novelty in the method reported in the manuscript.

We look forward to receiving your revised manuscript.

Kind regards,

Niraj Agarwala, Ph.D.

Academic Editor

PLOS ONE

Journal Requirements:

3. Thank you for stating the following in the Acknowledgments Section of your manuscript: "This work was supported by funds from the National Natural Science Foundation of China (81773844, 31000136) and the Beijing Municipal Natural Science Foundation (6102024). "

Please remove any funding-related text from the manuscript and let us know how you would like to update your Funding Statement. Currently, your Funding Statement reads as follows: "The author(s) received no specific funding for this work"

6. Please upload a new copy of Figure 9 as the detail is not clear. Please follow the link for more information: https://blogs.plos.org/plos/2019/06/looking-good-tips-for-creating-your-plos-figures-graphics/" https://blogs.plos.org/plos/2019/06/looking-good-tips-for-creating-your-plos-figures-graphics/

7. Please include your tables as part of your main manuscript and remove the individual files. Please note that supplementary tables (should remain/ be uploaded) as separate "supporting information" files

Reviewers' comments:

Reviewer's Responses to Questions

**Comments to the Author**

1. Is the manuscript technically sound, and do the data support the conclusions?

Reviewer #1: Partly

Reviewer #2: Partly

Reviewer #3: Partly

2. Has the statistical analysis been performed appropriately and rigorously? 

Reviewer #1: N/A

Reviewer #2: I Don't Know

Reviewer #3: No

3. Have the authors made all data underlying the findings in their manuscript fully available?

Reviewer #1: No

Reviewer #2: No

Reviewer #3: No

4. Is the manuscript presented in an intelligible fashion and written in standard English?

Reviewer #1: No

Reviewer #2: Yes

Reviewer #3: No

5. Review Comments to the Author

Reviewer #1: Dear authors, after a thorough revision of the manuscript titled “Fusarium oxysporum infection-induced formation of agarwood (FOIFA): a novel method for inducing the production of high-quality agarwood”, I would summarize my comments/recommendations as follows -

This study experiments with a particular isolate of Fox and validates its virulency and agarwood formation capacity. It is not clear from the MS why only a single Fox isolate was used in the study and the justifications for choosing the same. The role of other co-occurring microbes in the overall process of virulency and agarwood formation has not been discussed. It is apparent that the roles of those co-occurring microbes were completely ruled out during the experimentation part as well as MS written part. In this way, it is too early to say that only the said strain of Fox is responsible for better agarwood formation. Besides, the environment too plays a considerable effect in this regard, there is a lack of discussion on this too. In many places, the experimental details are not sufficient or not properly written to replicate the same. And the way in which the term “novel” is being used in places needs reconsiderations or justifications in the MS. These issues need to be addressed in the MS in order to make it complete.

1. The title “Fusarium oxysporum infection-induced formation of agarwood (FOIFA): a novel method for inducing the production of high-quality agarwood” mentions this as a novel method, while there are reports of Fusarium spp. induced agarwood formation “Example - Faizal, A., Esyanti, R.R., Aulianisa, E.N. et al. Formation of agarwood from Aquilaria malaccensis in response to inoculation of local strains of Fusarium solani. Trees 31, 189–197 (2017). https://doi.org/10.1007/s00468-016-1471-9”. Therefore, the authors should reconsider incorporating the term “novel” in the title of the MS.

2. Line 65-66 says that “in order to help meet the demand for agarwood production while protecting wild Aquilaria trees, this study had three objectives” but from the MS, it is nowhere discussed how the isolate reported in this study are protecting wild Aquilaria trees. There should be at least a few sentences in the discussion section regarding this.

3. Objective 1 in Line 66-67 says that “i) to identify fungal pathogens of Aquilaria that can induce the production of agarwood”, but the procedures and results describe only one Fusarium strain, and there is no justification why this particular isolate was used out of all the isolates.

4. Line 83: rewrite - “Sequence obtained were used as queries to search in Genbank”

5. Line 84: Mention why only F. oxysporum isolate (AsFo20150101) was used in this study, and give the BLAST results of all other isolates (which you have sequenced) in supplementary data.

6. Line 90-93: Mention the number of replica plants.

7. Line 95: Mention the exact time of harvesting stem tissues in dpi (day post-inoculation).

8. Line 99: “submitted to PCR” – mention which region/gene was amplified.

9. Line 101: The statement “The PCR products were checked using gel electrophoresis” is too vague to speculate what was done afterward. There should be a clear mention of whether the amplicon was sequenced or just visualized with control bands.

10. Line 112: “at 40 mg/ml” Rewrite as “at concentration 40 mg/ml”.

11. Line 117-118: The procedure should be briefly described.

12. Line 133: The conditions under which chromatography was carried out should be mentioned.

13. In Figure 2: The Fox isolate used in this study should be distinctly visualized from the remaining database isolates. Also, mention the strain number of the isolate (which you have assigned), and the accession number of the submitted sequence like those mentioned for database isolates in your phylogenetic diagram.

14. The discussion is a repetition of the results, making the content in this section very small. All the findings should be justified and critically discussed with reference to existing literature.

15. Line 247-249: “In this study, another novel technique for agarwood induction, known as the fungal agarwood inducing technique (Agar-Fit), was invented” – this statement is self-contradictory as earlier in the material method section (Line 93-94) it has been mentioned that the inoculation was done by following a method described by Zhang et al., 2014. Therefore, again reconsider using the term “novel” here.

16. Line 253-254: “the yield of agarwood can be further improved by combining biological inducers and chemical agents” – this sentence is too vague to state as there is no mention of an experiment in which the reported Fox strain was co-inoculated with chemical agents.

Reviewer #2: I have reviewed the paper thoroughly. There is merit in the findings that establishes Fusarium as a strong associate in agarwood -fungus ecology related to agarwood formation. The association of fungus in agarwood formation is a complex phenomenon which is still not clearly understood and has been known to involve several genera of filamentous fungi. The complex ecology also includes insects that probably create injuries for fungi to establish well. Studies that investigate this aspect are welcome as this has potential for artificial induction of agarwood in plantations and spare the natural trees of which only few are left under threat of extinction. But, it must be kept in mind at the same time that the association of Fusarium with agarwood is well known and since agarwood artificial infection is a commercially employed practice with high level of propriety and secrecy, it is quite likely that the same might be in use in formulations used by commercial agarwood producers already. At the same time, the study has a number of scientific and methodological shortcomings which need to be looked into extensively by the authors.

#1. The title needs revision (Ref: Line No. 1). The term FOIFA in line with AgarWit is misplaced since the paper does not establish it as a comprehensive technique for field inoculation. The passing mention is made only in the discussion and no comparative study of the two or different methods of infection has been provided to justify a new improved method.

#2. The information about the organism (Fusarium oxysporum strain) provided is very limited. The isolation, initial number of different isolates, screening from the lot and basis of selection are not presented. As per authors Fusarium solani has the most virulent association ((Ref: Line No. 62)

#3. Material and methods part needs a complete overhaul.

# 3.1. The experimental design for inoculation- how many plants, age of the plants, time period, frequency of success is missing. This should in fact have been clearly presented in a scientific manner and hence and detailed reports on it are required to make any proper assessment. (Ref: Line No. 90)

#3.2. Experimental details like selective media, primer sequence, amplicon size etc are missing (Ref: Line No. 97-101).

#3.3 The inoculation studies and pathogenicity test cannot be differentiated. Are they one and the same? Details are not clear ((Ref: Line No. 102-107).

# 3.4. An important reference resinous agarwood referred to as Wild agarwood (W) has been taken against which the present technique is compared. However its details particularly of source and nature are missing.

#3.5. The HPLC study selected 4 chromones. The basis for such selection is not clarified, source of sample, method of sample preparation are not discussed and source of standards not mentioned making the study weak. ((Ref: Line No. 120-128)

#3.6. The process of alcohol extract and essential oil being critical to the study require explanation and details. The authors have just mentioned a reference (Gao et al 2020). (Ref: Line No. 116-119)

#4. For the results provided I have the following observations:

#4.1. The data on isolation of fungi their screening etc is missing (as in #2)

#4.2. The results on the inoculation and its difference from pathogenicity study are missing (as raised in 3.1 to 3.4)

#4.3. TLC results ((Ref: Line No. 173-181) needs revision. Authors mentions “more constituents and higher contents…” which is not explained in Fig 5 or anywhere else.

#4.4. Chromone study I feel needs a complete revision. The points raised in 3.5 and in Fig 7, chromatogram B, C and D seem to tell a different story than what is mentioned in the manuscript particularly with regard to (if Fig 7D is W).

#4.5. GC results (Ref Table 1)indicate that a major sesquiterpene of agarwood agarospirol is undetected in control (W) which makes the choice of W as a reference for comparison unclear. Hence the claim of better sesquiterpene yield by the FOIFA method is confusing. The gain in B over W seems to be contributed mostly by high alpha selinene for which references are limited for agarwood. Please explain in detail.

#4.6. The figure 9 ABC legends are missing. So without knowing which chromatogram is what it is not possible to assess.

Reviewer #3: The technology used here is a very well known technology. it lacks new knowledge or science. Fu.ngal infection and physical methods and combitaion of both are well known see review CHHIPA and Kaushik ( 2017)

6. PLOS authors have the option to publish the peer review history of their article (what does this mean?). If published, this will include your full peer review and any attached files.

Reviewer #1: No

Reviewer #2: **Yes: **Supriyo Sen

Reviewer #3: **Yes: **Nutan Kaushik

---

## [Author Response · Author response to Decision Letter 0]

7 Jul 2022

Response to reviewer #1: Dear reviewer, we appreciate for your valuable suggestions on the problems existing in the article. After reviewing your comments carefully, the following are my responses to your suggestions：

1. The title: did reports Fusarium oxysporum can induce the formation of agarwood, and we study on the basis of justifying the formation of the new technology to rapid and efficient high-quality aloes, and as a result, we are also in the title changed, more prominent we validate the technology of rapid and efficient, in order to more relevant to our research.

2. Line 309-318: We added this part in the discussion about how the isolate reported in this study are protecting wild Aquilaria trees，this was indeed not shown in the previous content, thank you for your advice.

3. Line 49-59: We mention that many kinds of extracts have been obtained from natural agarwood, and the toxicity and induced of these extracts to agarwood have been reported，and Fusarium oxysporum is highly virulent to Aquilaria was known, so that we chose this particular isolate for our study

4. Line 78: We have corrected the mistake, thanks for your suggestion.

5. Line 79: In the research process, we focused on the influence of Fusarium oxysporum on the quality of agarwood, and did not discuss other strains in depth. We think this is not the main research direction of this paper, which is also our follow-up research target.

6. Line 88-99: According to your advice, we have added the specific description of sample handling and improved the relevant information.

7. Line 88-99:As above, we have added relevant content.

8. Line 105-108: Amplified gene information has been supplemented.

9. Line 109: We have clarified the subsequent verification process in the article. 

10. Line 122: Your comments are very detailed, and we have revised them. Thank you.

11. Line 128-140: We have supplemented the specific operation steps, which is very important for the integrity of our research.

12. At line 160, we have added corresponding contents. Thank you for your suggestions.

13. Figure 2: According to your suggestion, we have modified the picture in order to fully reflect the relevant information.

14. Line 264: The discussion part of the article is really limited, and we also hope that it can be helpful to relevant research without causing trouble, so we have added some justified and critically discussed as you suggested.

15. Line 286: As you mentioned earlier, the term "Novel" really does not capture the focus of our research. Therefore, we have reconsidered this aspect and thank you again for your preciseness

16. Line 297: We have corrected this statement to make it clearer and more valuable for reference. Thank you for your advice.

We appreciate again for your meticulous suggestions on this study, so that our paper has been better supplemented and modified. We look forward to your reply to our revised draft again, thank you! Best wishes!

Response to reviewer #2: Dear reviewer, Thank you very much for your affirmation and recognition of our research. We know that this technology may be of great value in production application, so we have made corresponding changes to your suggestions. 

1. As for the title (line 1), in this paper, we focus on FOIFA this novel technology for the influence of the quality of aloes, also found that the technique can really improve the production of high quality aloes, so we think that the technology in the production of artificial aloes with innovative significance and application value, which is a kind of improvement of production methods of artificial agarwood.

2. At line 67, according to your suggestion, we have carried out detailed description and information provision for the isolated strains to ensure the reliability and credibility of the strains we used, which is very important for this technology. Your opinion also ensures the integrity of the method.

3. As for materials and methods, our description of some details is indeed insufficient, and we also refer to a lot of relevant literature published by the laboratory before, without considering a complete description of all materials and methods in this paper. In this revised draft, We have clearly explained the details of plants (line 86), genes (line 76), samples (line 88), HPLC (line 142) and extraction process (line 128) you mentioned, and hope the supplementary content can answer your doubts.

4. As for the results, as you mentioned, we have supplemented some missing key information data. Such as fungus in the process of separation of the screening data were added in the method (line 67), the data and results to organize and expounded, and the results of GC to complete interpretation (Table 1), indeed can explain the increase of the ratio W B is aloes quality enhancement, but also changed the image of legend, to complete our results, these have been revised in the manuscript, Thank you very much for asking these questions to ensure the integrity of the article.

We appreciate again for your meticulous suggestions on this study, so that our paper has been better supplemented and modified. We look forward to your reply to our revised draft again, thank you! Best wishes!

Response to reviewer #3: Dear reviewer, in this paper, we focus on FOIFA this novel technology for the influence of the quality of aloes, also found that the technique can really improve the production of high quality aloes, so we think that the technology in the production of artificial aloes with innovative significance and application value, which is a kind of improvement of production methods of artificial agarwood. Thanks a lot! Best wishes!

---

## [Decision Letter · Decision Letter 1]

11 Aug 2022

PONE-D-22-01806R1Fusarium oxysporum infection-induced formation of agarwood (FOIFA): a rapid and novel efficient method for inducing the production of high quality agarwoodPLOS ONE

Dear Dr. xiangzhao,

Thank you for submitting your manuscript to PLOS ONE. After careful consideration, we feel that it has merit but does not fully meet PLOS ONE’s publication criteria as it currently stands. Therefore, we invite you to submit a revised version of the manuscript that addresses the points raised during the review process.

We look forward to receiving your revised manuscript.

Kind regards,

Niraj Agarwala, Ph.D.

Academic Editor

PLOS ONE

Journal Requirements:

Reviewers' comments:

Reviewer's Responses to Questions

**Comments to the Author**

1. If the authors have adequately addressed your comments raised in a previous round of review and you feel that this manuscript is now acceptable for publication, you may indicate that here to bypass the “Comments to the Author” section, enter your conflict of interest statement in the “Confidential to Editor” section, and submit your "Accept" recommendation.

Reviewer #1: All comments have been addressed

Reviewer #2: All comments have been addressed

Reviewer #3: All comments have been addressed

2. Is the manuscript technically sound, and do the data support the conclusions?

Reviewer #1: Yes

Reviewer #2: Yes

Reviewer #3: Yes

3. Has the statistical analysis been performed appropriately and rigorously? 

Reviewer #1: N/A

Reviewer #2: (No Response)

Reviewer #3: Yes

4. Have the authors made all data underlying the findings in their manuscript fully available?

Reviewer #1: Yes

Reviewer #2: (No Response)

Reviewer #3: Yes

5. Is the manuscript presented in an intelligible fashion and written in standard English?

Reviewer #1: No

Reviewer #2: Yes

Reviewer #3: Yes

6. Review Comments to the Author

Reviewer #1: Dear authors, the updated version of the manuscript has been improved in many ways than the previous version. However, there are many aspects in the manuscript where I find the facts presented a bit rimy and superficial. I also feel a lack of general veracity as several of my comments has been deliberately ignored or misunderstood. In general, would like to put my comments as follows:

1. Line 79 – 80: From the description of the subsection “Isolation and identification of fungal isolates” under the section “materials and methods” it is clear that many fungal isolates were isolated from surface-sterilized symptomatic Aquilaria sinensis trees. Among all the isolates the F. oxysporum isolate (AsFo20150101) was selected for the study. But it has not been properly justified why among all, this particular isolate was chosen. The description you are citing (Line 49 – 59) does not justify that. Mention the reason why other isolates were ruled out for the study and why isolate AsFo20150101 was kept; that will suffice.

2. In Figure 2: The accession number of the Fox isolate used in the study is still missing in the phylogenetic tree. It should be mentioned.

3. The discussion section is still not adequate as per the journal requirement. There are many earlier reports where Fox isolates have been reported to induce agarwood formation, where the underlying mechanisms and factors have been deciphered. The discussion section of this manuscript should draw the connections from earlier studies and justify the (possible) mechanisms/roles of the AsFo20150101 for the results shown in the current investigation.

With these modifications, the manuscript will be complete.

Reviewer #2: The Title "novel" should be replaced by "efficient" ; presently both words appear complicating the readability of the title

Reviewer #3: they addressesd all issues. manuscript may be accepted. agarwood research needs more attention as its larg scale application is limited

7. PLOS authors have the option to publish the peer review history of their article (what does this mean?). If published, this will include your full peer review and any attached files.

Reviewer #1: No

Reviewer #2: No

Reviewer #3: **Yes: **Nutan Kaushik

---

## [Author Response · Author response to Decision Letter 1]

23 Sep 2022

Response to reviewer #1: Dear reviewer, thank you for your careful review of our manuscript, and we are very sorry for some of the unintentional mistakes we caused in responding to your previous comments. We have carefully made the following changes to your comments:

1.Line 79-80: As you have said, there are many fungal isolates that we has isolated from surface-sterilized symptomatic Aquilaria sinensis trees. But unfortunately, after our preliminary screening, only the F. oxysporum isolate (AsFo20150101) showed significant effect, which is why our research revolves around this effective isolate,we also made modification in the article to eliminate the misunderstanding.

2.In Figure 2: We have inserted the accession number of the F. oxysporum isolate used in the study(MW880244.1) into the phylogenetic tree, we apologize for not noticing this in the previous revision.

3.Indeed, we did not thoroughly explore the potential mechanism of agarwood formation induced by Fusarium before. Therefore, we combined with the previous study to explore the possible mechanism of the F. oxysporum isolate for the results shown in the current investigation, hoping to supplement the missing content.

We appreciate again for your meticulous suggestions on this study, so that our paper has been better supplemented and modified. We look forward to your reply to our revised manuscript again, thank you! Best wishes!

Response to reviewer #2: Dear reviewer, thank you for your suggestion about the title modification, which makes our title more readable and easy to understand. Thank you again for your suggestion and wish you a happy life.

Response to reviewer #3: Dear reviewer, thank you for your affirmation of our research and work. We also hope that our research can be of some help to the application of agarwood. Thanks again and wish you happiness

---

## [Decision Letter · Decision Letter 2]

21 Oct 2022

Fusarium oxysporum infection-induced formation of agarwood (FOIFA): a  rapid and efficient method for inducing the production of high quality agarwood

PONE-D-22-01806R2

Dear Dr. xiangzhao,

We’re pleased to inform you that your manuscript has been judged scientifically suitable for publication and will be formally accepted for publication once it meets all outstanding technical requirements.

Kind regards,

Niraj Agarwala, Ph.D.

Academic Editor

PLOS ONE

Additional Editor Comments (optional):

Reviewers' comments:

Reviewer's Responses to Questions

**Comments to the Author**

1. If the authors have adequately addressed your comments raised in a previous round of review and you feel that this manuscript is now acceptable for publication, you may indicate that here to bypass the “Comments to the Author” section, enter your conflict of interest statement in the “Confidential to Editor” section, and submit your "Accept" recommendation.

Reviewer #1: All comments have been addressed

Reviewer #2: All comments have been addressed

2. Is the manuscript technically sound, and do the data support the conclusions?

Reviewer #1: Yes

Reviewer #2: Yes

3. Has the statistical analysis been performed appropriately and rigorously? 

Reviewer #1: N/A

Reviewer #2: Yes

4. Have the authors made all data underlying the findings in their manuscript fully available?

Reviewer #1: Yes

Reviewer #2: (No Response)

5. Is the manuscript presented in an intelligible fashion and written in standard English?

Reviewer #1: Yes

Reviewer #2: Yes

6. Review Comments to the Author

Reviewer #1: Dear authors, all the issues have been addressed properly and formatted as per the journal requirements. Therefore, the manuscript is ready now.

Reviewer #2: (No Response)

7. PLOS authors have the option to publish the peer review history of their article (what does this mean?). If published, this will include your full peer review and any attached files.

Reviewer #1: No

Reviewer #2: **Yes: **Supriyo Sen
